# OmniBal: Towards Fast Instruction-Tuning for Vision-Language Models via Omniverse Computation Balance

Yongqiang Yao [* 1]   Jingru Tan [* 2]   Feizhao Zhang [* 3]   Jiahao Hu [3]   Yazhe Niu [4]   Xin Jin [3]   Bo Li [5]   Pengfei Liu [1]
Ruihao Gong [6 3]   Dahua Lin [3 4]   Ningyi Xu [1]

## Abstract

Vision-language instruction-tuning models have recently achieved significant performance improvements. In this work, we discover that large-scale 3D parallel training on those models leads to an imbalanced computation load across different devices. The vision and language parts are inherently heterogeneous: their data distribution and model architecture differ significantly, which affects distributed training efficiency. To address this issue, we rebalance the computational load from data, model, and memory perspectives, achieving more balanced computation across devices. Specifically, for the data, instances are grouped into new balanced mini-batches within and across devices. A search-based method is employed for the model to achieve a more balanced partitioning. For memory optimization, we adaptively adjust the re-computation strategy for each partition to utilize the available memory fully. These three perspectives are not independent but are closely connected, forming an omniverse balanced training framework. Extensive experiments are conducted to validate the effectiveness of our method. Compared with the open-source training code of InternVL-Chat, training time is reduced greatly, achieving about $1.8\times$ speed-up. Our method's efficacy and generalizability are further validated across various models and datasets. Codes will be released at https://github.com/ModelTC/OmniBal.

## 1. Introduction

Large language models (LLMs) have brought new possibilities to many fields. Multi-modal models, particularly Vision-Language Models (VLMs) (Alayrac et al., 2022; Team et al., 2023a; Reid et al., 2024; Liu et al., 2023a; Bai et al., 2023b; Chen et al., 2023), are advancing rapidly due to their deeper understanding of the world. The training scale of Vision-Language Models (VLMs) continues to expand, with increasingly larger datasets incorporating more text and higher-resolution images. Compared with the LLaVA-1.5 (Liu et al., 2023a), the InternVL-Chat (Chen et al., 2024) has expanded the dataset size from 665K to 5M and increased image resolution from 336x336 to 3840x2160. At the model level, larger vision encoders are adopted. The InternVL-Chat upgrades the visual encoder from ∼300M ViT-L-336px (Radford et al., 2021) to ∼6B InternViT-448px (Chen et al., 2023).

The larger datasets and models result in a more time-consuming training process. Therefore, efficient training strategies are essential for the rapid advancement of the field. 3D parallelism (Shoeybi et al., 2019; Rajbhandari et al., 2020; microsoft, 2020) is a popular framework for large-scale distributed training, which allows data and models to be distributed across multiple devices. Balancing computational load across devices is crucial in 3D parallelism by minimizing idle times.

In this work, we find that for instruction-tuning large vision-language models, the heterogeneous nature of data and model structures brings new challenges to 3D parallelism training: (1) Varying input sizes of LLM and VIT cause imbalanced computational loads across training iterations and devices. (2) The heterogeneity between LLM and VIT models leads to inherent differences in the computational load of their transformer blocks. Along with varying input sizes, this inevitably results in uneven computational load and computational bubbles. (3) Input size variation and computational imbalance compel us to use the most aggressive re-computation (checkpointing) (Li et al., 2014) strategy to prevent program crashes, which wastes computational resources. We refer to those issues caused by the heterogeneity in data and model structures in large vision-

---
[*]Equal contribution  [1]Shanghai Jiao Tong University [2]Central South University [3]SenseTime Research [4]The Chinese University of Hong Kong [5]Tongji University [6]Beihang University. Correspondence to: Ruihao Gong <gongruihao@buaa.edu.cn>, Ningyi Xu <xuningyi@sjtu.edu.cn>.

*Proceedings of the $42^{nd}$ International Conference on Machine Learning*, Vancouver, Canada. PMLR 267, 2025. Copyright 2025 by the author(s).

language models as the **Computation Imbalance** problem, which reduces training efficiency.

To address this problem, a simple and efficient training framework called Omniverse Balance (**OmniBal**) is proposed to balance computational load across multiple devices. This framework systematically balances computation in three bottlenecks, *i.e.* data, model, and memory, as shown in Figure 1. OmniBal works in these three closely connected aspects. Data lays the groundwork for addressing model imbalances, while data and model form the foundation for solving memory issues. Ultimately, these three aspects collaborate to achieve balanced computation. **Data:** The balanced dynamic mini-batch method is proposed to group instances as new mini-batches according to text length and number of images. Specifically, an iterative algorithm based on sampling and filtering combines data of different sizes into balanced groups, ensuring stable input sizes; **Model:** We propose balanced model partitioning to evenly spread the computational load of LLM and VIT across devices. Using a search-based approach, we efficiently find optimal partition strategies within a small search space, enabling adaptation to different model architectures and hardware platforms. The balanced dynamic mini-batch method facilitates balanced model partitioning by ensuring input sizes are consistent in advance. **Memory:** A balanced adaptive re-computation method is proposed to optimize the re-computation strategy on each device, maximizing both memory utilization and training speed. We calculate the memory requirements of different models to adjust the re-computation strategy adaptively. Notably, our proposed balanced dynamic mini-batch and model partitioning ensures balanced computational loads on each device, making memory analysis feasible.

Extensive experiments are performed on various open-source VLM models at different scales, reducing overall training times significantly. GPU days are reduced for InternVL-Chat-1.5 (6+20B) from 61.8 to 21.3 under the Megatron-DeepSpeed (microsoft, 2020) backend. Scaling up to InternVL-Chat-1.5-Plus (6+34B), we consistently observe a great speed-up, from 75.4 to 30.5 GPU days. We conduct thorough generalization experiments, including various datasets, hardware configurations, and multiple model combinations. Consistent and substantial improvements are observed across all experiments, demonstrating the effectiveness and versatility of our method.

In summary, our contributions are reflected in three levels: framework, method, and results.

- We are the first to identify and address the computational imbalance problem in large-scale VLM (Vision-Language Model) instruction-tuning training, proposing a systematic solution tailored to this challenge.

- Our approach is systematic rather than isolated and incremental, integrating the three interconnected modules of data, model, and memory to ensure computational balance across both inter- and intra-stage levels.

- Our method achieves a great speed-up while maintaining model performance on the open-source training code. Its efficacy and generalizability are further validated across various models datasets and tasks.

## 2. Related Works

### 2.1. Multi-Modal Large Language Model(MLLM)

Large language models, such as ChatGPT (OpenAI, 2023a), GPT-4 (OpenAI, 2023b), Llama series (Touvron et al., 2023a;b; AI, 2024), and Gemini series (Team et al., 2023b; Reid et al., 2024), have seen significant advancements recently. They rely on large datasets for training to achieve strong performance, particularly in few-shot and zero-shot scenarios. Typically, they are built on textual data and can only accept text inputs. However, real-world scenarios often involve rich multi-modal information, *e.g.*, images. It has driven the development of large vision language models (VLMs). Visual encoders like Vision Transformer (ViT) (Dosovitskiy et al., 2021) usually incorporate vision information. A cross-modal connector is also required to align the vision encoder outputs to the language models. LLaVA (Touvron et al., 2023a) uses the simplest MLP, BLIP series (Li et al., 2022; 2023; Dai et al., 2024) uses Q-former, Qwen-VL-Chat (Bai et al., 2023b) uses a cross-attention module. VLMs expand large language models' capabilities and application scenarios by instruction-tuning with text and image data. However, introducing multi-modal data and heterogeneous encoders brings challenges to the training.

### 2.2. Large-Scale Distributed Training

Distributed training is essential for efficiently utilizing multiple GPUs to train large language models. It is achieved through 3D parallelism (Shoeybi et al., 2019; Rajbhandari et al., 2020; microsoft, 2020): data, tensor, and pipeline parallelism. *Data Parallelism* splits the entire dataset into mini-batches and assigns them to multiple devices, each with a model replica. This approach maximizes the use of GPU power for large datasets. DeepSpeed Zero (Rajbhandari et al., 2020) enhances it by reducing weight redundancy. However, it can still be challenged by the memory limits of individual devices when handling huge models. *Tensor Parallelism* distributes a model's weight matrices across multiple devices, enabling parallel matrix operations (Shoeybi et al., 2019) and reducing per-device memory requirements. This method accelerates computation but requires dense inter-device communication, typically restricted to single-node deployments to minimize latency. *Pipeline Parallelism*

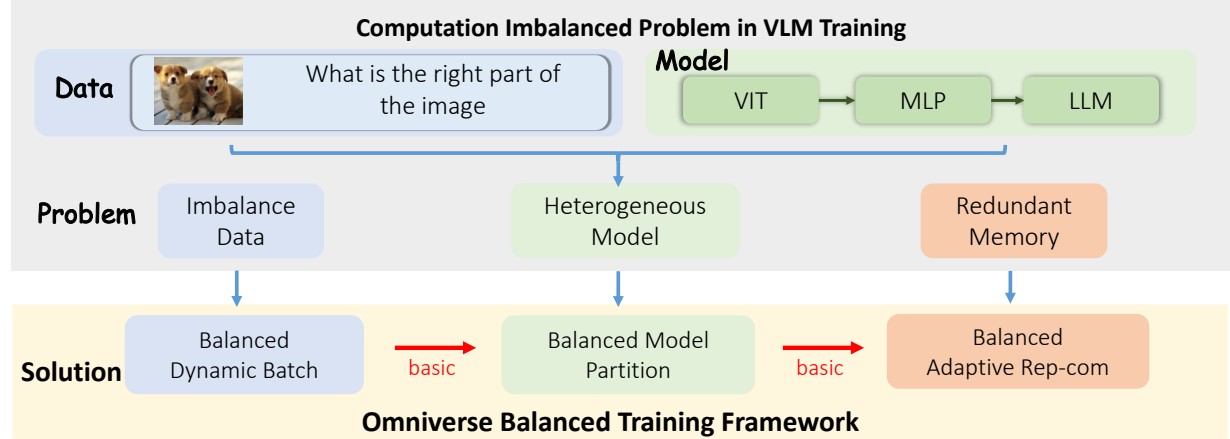

*Figure 1.* Overview of the computation imbalanced problem and our proposed solution in the Standard Vision-Language instruction-tuning framework. We consider the bottleneck issues of data, model, and memory, and propose an omniverse solution addressing these three aspects, each providing the foundation for the next.

*Table 1.* Analysis of computation imbalance.

| Dimension | Input | Forward time | Cost memory |
|---|---|---|---|
| Inter-Stage | $1.4K \pm 0.9K token$ | $85 \pm 93$ ms | $39 \pm 23$ G |
| Intra-Stage-1 | $1.9K \pm 1.2K token$ | $136 \pm 155$ sm | $73 \pm 6$ G |

divides a model into segments and assigns them to different devices, creating a computation flow like a production line. This technique facilitates larger model scaling across nodes. GPipe (Huang et al., 2019) proposes micro-batching to decrease forward bubbles. PipeDream (Narayanan et al., 2019) further proposes a one-forward-one-backward (1F1B) scheme to optimize memory usage. In pipeline parallelism, uneven layer partitioning can cause significant pipeline bubbles. PipeDream (Narayanan et al., 2019) and AdaPipe (Sun et al., 2024) optimize model partitioning and re-computation strategies based on profiling and dynamic programming, respectively. However, these advancements are primarily tested in text-based models and may require adaptation for large vision-language model scenarios.

## 3. Computation Imbalance

In this section, we explore the unique challenges of large-scale distributed training for vision-language models, focusing on two dimensions: **Inter-Stage** and **Intra-Stage** computation imbalance. Inter-Stage means the computation imbalance of different pipeline parallel stages. Intra-Stage indicates the computation imbalance of the same stage across time and devices. Figure 2 shows these two computation imbalances more intuitively. And they both include three specific levels: data, model, and memory. To quantify this problem, we used the InternVL-Chat-1.2 dataset (Chen et al., 2024) to perform profile statistics shown in Table 1. For the Intra-Stage, we counted the information of Stage 1

as a sample.

**Data Imbalance:** LLMs are trained on texts using next-token prediction, allowing consistent input lengths through arbitrary text sub-strings. In contrast, VLMs handle texts and images, requiring data integrity, and preventing arbitrary truncation. The varying number of images, resolutions, and text lengths result in considerable differences in input sizes across mini-batches. From Table 1 and Figure 2, data imbalance occurs in Inter-Stage and Intra-Stage. To better quantify the impact of dynamic input, we define the DistRatio (introduced in Section 4) to measure the degree of data imbalance of VIT and LLM.

**Data Imbalance Evidence:** As shown in Figure 2, at time T-0, the Vision and LLM inputs for DP-0 (Group 0 of data-parallel) and DP-1 (Group 1 of data-parallel) differ (Img=4, Text=2k vs. Img=9, Text=4k). Different inputs will bring different computational complexities. For example, the total forward time of DP-0 at time T-0 is 970ms (500 + 160 + 150 + 160), and the total forward time of DP-1 at time T-0 is 1730ms (900 + 300 + 260 + 270). DP-0 needs to wait for DP-1 to complete all training before updating parameters, which will cause DP-0's GPU resources to wait for a long time. This is the Intra-Stage imbalance problem across devices. For DP-0, the inputs at times T-0 and T-1 on the same device also differ significantly (e.g., Img=4, Text=2k versus Img=20, Text=16k), which is an Intra-Stage imbalance problem over time. This results in a substantial variance in forward time and memory usage, creating challenges in determining a globally optimal model partition strategy.

**Model Imbalance:** LLMs use identical transformer modules with the same computational load. Evenly dividing these layers in pipeline parallelism distributes the load effectively. However, VLMs require additional image pre-

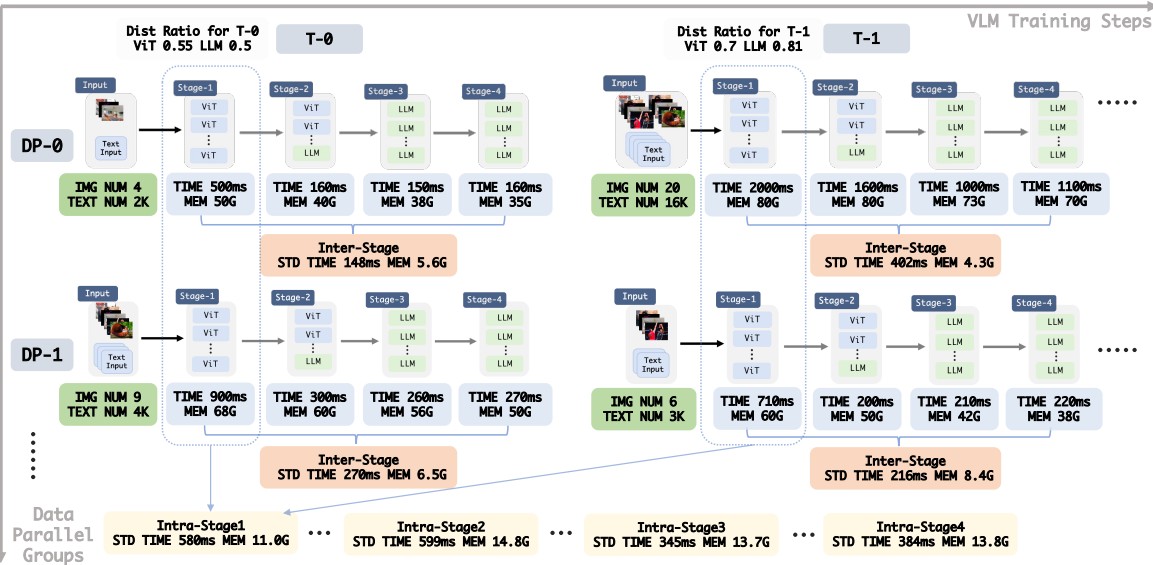

*Figure 2.* The Problem of Computation Imbalance in VLM Instruction-Tuning Training Pipeline. DP-0 and DP-1 represent different Data Parallel processes. T-0 and T-1 represent different training times. TIME and MEM represent forward time and cost memory in the current stage, respectively. STD stands for standard deviation.

processing, necessitating an image encoder. The structural disparity between VIT and LLM results in different computational demands. From Table 1 and Figure 2, the standard deviation of forward time is huge in both Inter-Stage and Intra-Stage, indicating a serious computation imbalance.

**Model Imbalance Evidence:** As shown in Figure 2, for DP-0, at T-0 and T-1, the standard deviation (148ms and 402 ms) of the forward time across different stages is significant, leading to serious bubbles in pipeline parallelism and slowing down training speed. This is the Inter-Stage imbalance problem. Furthermore, the computation distribution across stages shows a huge difference between T-0 and T-1. For example, the computation distribution at T-0 is (500:160:150:160)=(0.52:0.16:0.15:0.16),while the computation distribution at T-1 is (2000:1600:1000:1100)=(0.35,0.28,0.18,0.19).The varying computation distribution makes determining the optimal model partition strategy challenging. This includes the Inter-Stage and Intra-Stage problems (over time).

**Memory Imbalance:** LLMs require significant GPU memory due to their large parameter size. When memory is insufficient, re-computation (Li et al., 2014) techniques discard some intermediate activation values and recompute them during backward propagation to save memory. VLM encounters great memory challenges due to the variable scales of data inputs and the heterogeneity between vision and language models. The presence of numerous images or long text inputs can lead to excessive GPU memory usage, requiring the most aggressive re-computation settings

to prevent the program from crashing. However, excessive re-computation can slow down the training process. From Table 1 and Figure 2, memory imbalance is reflected in both Inter-Stage and Intra-Stage in the existing training setting.

**Memory Imbalance Evidence:** As shown in Figure 2, the memory consumption for Stage-1 at different time points is as follows: 50 GB at T-0, 80 GB at T-1 for DP-0, 68 GB at T-0, and 60 GB at T-1 for DP-1. To prevent program crashes, we must configure the re-computation strategy based on the highest memory usage, which is 80 GB. This approach introduces additional computational overhead, categorized as the Intra-Stage imbalance problem (over time). Additionally, for DP-1, the memory usage across different stages at both T-0 and T-1 has a high standard deviation (6.5 GB and 8.4 GB, respectively). Using the same re-computation strategy for different stages will also bring computational overhead, according to the highest memory usage. This variation represents the Inter-Stage imbalance problem.

**Differences between VLM and LLM training**: As mentioned above, the difference between VLM and LLM arises from the data composition and model structure, resulting in unique Inter-Stage and Intra-Stage challenges. Inter-Stage: Since LLM has a fixed structure, the model can be equally divided, and there is no Inter-Stage imbalance for any input. Dynamic input or inconsistent text-image ratios in heterogeneous VLM will lead to an Inter-Stage imbalance problem. Intra-Stage: For the LLM-Pretrain task, the input is fixed, and there is no Intra-Stage imbalance problem. Dynamic input can be converted into static input by simple packing

([Kosec et al., 2021](#)) to reduce the computation imbalance for the LLM-SFT task. However, VLM instruction-tuning training cannot rely on simple packing to ensure fixed inputs for VIT and LLM, resulting in computation imbalance problems.

# 4. Method

This section presents our computation-balanced framework, OmniBal, for training large vision-language models. To address an imbalanced computational load across devices, we first manage the large data input variations, which is the most fundamental issue in the computation imbalance problem. This enables balanced model partitioning. Finally, the re-computation strategy for each partition is optimized. Appendix A shows our training pipeline.

---

**Algorithm 1** ISF: Iterative Sampling and Filtering

---
1: **ISF: Sampling Stage**
2: $\mathcal{D} = \text{randperm}(\mathcal{D})$, set $\mathcal{G} = [\ ]$
3: **for** $(x_i, y_i)$ in $\mathcal{D}$ **do**
4:     $\mathcal{G} \leftarrow \mathcal{G} + (x_i, y_i)$
5:     **if** $I_v > Q_v$ or $I_t > Q_t$ **then**
6:         $\mathcal{G} \leftarrow \mathcal{G} - (x_i, y_i)$
7:         $\mathcal{P} \leftarrow \mathcal{P} + \mathcal{G}$ , set $\mathcal{G} = [(x_i, y_i),]$
8:     **end if**
9: **end for**
10: **return**: $\mathcal{P}$

11: **ISF: Filtering Stage**
12: Get $\mathcal{P}$ from Sampling Stage
13: **for** $\mathcal{G}$ in $\mathcal{P}$ **do**
14:     **if** $I_v < Q'_v$ and $I_t < Q'_t$ **then**
15:         $\mathcal{P} \leftarrow \mathcal{P} - \mathcal{G}$
16:     **else**
17:         remove all $(x_i, y_i)$ of $\mathcal{G}$ from $\mathcal{D}$
18:     **end if**
19: **end for**
20: **return** $\mathcal{P}, \mathcal{D}$

---

## 4.1. Balanced Dynamic Mini-Batch

For instruction-tuning VLMs, each training sample contains various images and texts, resulting in non-fixed input sizes. We evaluate data imbalance from two perspectives: within-device samples and cross-device mini-batches.

**Pad Ratio (within-device):** When combining samples of different sizes into a mini-batch, smaller samples need to be padded to ensure aligned input sizes. The Pad Ratio is calculated as follows:

$$PadRatio = \frac{\sum_i^B (t_{max} - t_i)}{t_{max} \times B} \quad (1)$$

Where $t_{max}$ represents the maximum number of tokens in a mini-batch of size $B$, and $t_i$ denotes the number of tokens for sample $i$ within that mini-batch.

**Distribution Ratio (cross-device):** Even after padding, the sizes of mini-batches on different devices may vary, leading to different input scales across devices. The distribution ratio is calculated as follows:

$$DistRaito = \frac{\sum_i^N (T_{max} - T_i)}{T_{max} \times N} \quad (2)$$

Where $N$ represents the number of devices, $T_{max}$ denotes the maximum number of mini-batch tokens across all devices, and $T_i$ refers to the number of mini-batch tokens on the $i^{th}$ device. Non-fixed input sizes in VLMs have a larger Pad Ratio and Dist Ratio, as shown in Table 3 (row 1). A high Pad Ratio wastes computational resources, while a high Dist Ratio causes device idle time. They significantly impact training throughput efficiency.

To address this issue, an adaptive grouping strategy that organizes multiple samples, ensuring that both image and text sizes in the resulting groups remain within a relatively fixed range is implemented. We refer to this method as the Balanced Dynamic Mini-Batch. Determining the optimal grouping strategy is a non-trivial problem, An iterative method is designed using sampling and filtering to group samples. As illustrated in Algorithm 1, it **Iterative Sampling and Filtering (ISF)** involves the following steps:

*1.Sampling Stage:* For current dataset $\mathcal{D} = \{(x_i, y_i) \mid i\}$, we randomly add samples $d_i$ consisted of images $x_i$, text $y_i$ to current group $\mathcal{G}$. If the total number of images $I_v = \sum_{x_i \in \mathcal{G}} |x_i|$ or the total text length $I_t = \sum_{y_i \in \mathcal{G}} |y_i|$ reaches the predefined maximum number of images $Q_v$ or text $Q_t$, we add this group to the candidate set $\mathcal{P}$ and create a new group containing $(x_i, y_i)$ for the subsequent samples. Otherwise, we will continue adding samples to the current group. At the end of the sampling stage, we will have a candidate set $\mathcal{P} = \{\mathcal{G}_i | i = 1, 2, 3..\}$.

*2.Filtering Stage:* We first define the target number of images $Q'_v$ and text $Q'_t$. For each group $\mathcal{G}_i$ in candidate set $\mathcal{P}$, we keep $\mathcal{G}$ whose image number $I_v$ or text length $I_t$ satisfy $I_v >= Q'_v$ or $I_t >= Q'_t$, and remove all samples $(x_i, y_i)$ in that group from $\mathcal{D}$. Otherwise, we remove non-satisfied $\mathcal{G}_i$ from $\mathcal{P}$. Ultimately, $\mathcal{P}$ becomes our target set, and $\mathcal{D}$ becomes our updated dataset for the next iteration.

The sampling and filtering stages are alternately repeated for a maximum of $T$ times. The candidate set is acquired $\mathcal{P}$ each time, which includes more valid sample groups $\mathcal{G}$.

Meanwhile, we have the updated dataset $\mathcal{D}$ consisting of unselected samples, which is used for sampling and filtering in the next iteration. To ensure that the mini-batches constructed by the ISF method achieve lower Pad Ratio and Dist ratio, appropriate values for $Q_v$ and text $Q_t$ need to be determined. The optimal values for $Q_v$ and $Q_t$ vary across different datasets. In practice, A statistical approach described in Section 5.1 is used to determine these values.

## 4.2. Balanced Model Partitioning

Given the number of layers $L$ in the model and the pipeline parallel size $N$, our goal is to find an optimal partition strategy $P = (P^{(1)}, P^{(2)}, P^{(3)}, \ldots, P^{(N-1)})$ such that the training speed of the model is maximized. Here, $P^{(1)} < P^{(2)} < P^{(3)} < \ldots < P^{(N-1)}$, and the $i^{th}$ partition stage $S_i$ consists of layers $l_k$, where $P^{(i-1)} \leq l_k < P^{(i)}$, with $P^{(0)} = 1$ and $P^{(N)} = l + 1$. For example, given a model with $L = 20$ layers and pipeline size $N = 4$, assume that we have an optimal partition $P = (5, 10, 15)$. The first partition $S_i$ consists of layers $l_1, l_2, l_3, l_4$ since $P^{(0)} = 1, P^{(1)} = 5$.

However, achieving balanced pipeline partitioning for VLMs is a more challenging task compared to LLMs. We must consider: *(1) Model Heterogeneity:* The structural differences between visual and language models make simple parameter-based or layer-based partition strategies ineffective. *(2) Communication Overheads:* Different partitioning strategies result in varying communication volumes, as the number of activations in each layer can differ significantly in VLMs. *(3) Hardware Variability:* Different platforms exhibit varying levels of capability, particularly in terms of communication overhead. On platforms with high network bandwidth, communication overhead can be negligible. Based on the above analysis, A heuristic search algorithm to find the optimal partition is developed. We first identify a candidate set of partition strategies $\{P_k = (P_k^{(1)}, P_k^{(2)}, P_k^{(3)}, \ldots, P_k^{(N-1)}) \mid k = 1, 2, 3, \ldots\}$ that possibly contain the optimal one. Then, the optimal partition strategy $P^*$ is selected based on the running time:

$$P^* = \operatorname{argmin}_{P_i} f(P_i) \qquad (3)$$

Here, $f(P_i)$ is the average running time obtained by training the model for several iterations.

**Partition Candidates:** We start by profiling each layer's computation time $\text{FWD}(l_i)$. A greedy algorithm is employed to compute the anchor partition strategy $P^+$, making the computation time of all partition stages $S_i$ close. Around $P^+$, A candidate set of partition strategies is created by jittering $P^{(1)}, P^{(2)}, \ldots, P^{(N-1)}$ within a radius of $r$. When $r = 1$ and $N = 4$, there are a total of $3^3 = 27$ candidates.

**Partition Metrics:** When $r$ and $N$ are very large, there will be a vast number of partition candidates, making it inefficient to evaluate the running time for each one. Therefore, two metrics to rank these candidates are designed.

The first metric is the difference in running time between different pipeline stages $S_i$. Smaller differences generally result in fewer bubbles and faster execution. We use the variance of the running times of different pipeline stages to measure this difference.

$$\text{VAR(fwd\_time)} = \sum_{i=1}^{N} (\text{FWD}(S_i) - \overline{\text{FWD}(S_i)})^2 \qquad (4)$$

The second metric is the total point-to-point communication volume of the partition strategy $P_i$. It depends on $P_i$ consisting of $(P^{(1)}, P^{(2)}, P^{(3)}, \ldots)$

$$\text{SUM(comm)} = \sum_{i=1}^{N-1} \text{ACTIV}(l_{pi}) \qquad (5)$$

Where $l_{pi}$ is the last layer of partition strategy $P^{(i)}$ and $\text{ACTIV}(l_{pi})$ is the activation number of layer $l_{pi}$, indicating the point-to-point communication volume of $P^{(i)}$. We use the sum of VAR(fwd_time) and SUM(comm) as the metric for the partition and rank them to select the top $K$ candidates for speed evaluation.

## 4.3. Balanced Adaptive Re-Computation

Thanks to the balanced dynamic mini-batch and balanced model partition, a balanced computational load is maintained across each pipeline stage. The memory requirements are now stabilized as the computational demand has been fixed. As a result, we can optimize the re-computation strategy based on actual memory needs, rather than relying on the most aggressive approach to avoid crashes. Reducing the number of re-computations accelerates the model's backward pass, leading to faster training speed.

We find that heterogeneous architectures have different memory requirements. For example, the vision model in InternVL-Chat-1.5 requires more GPU memory than the language model under the same computational load. Therefore, it is necessary to analyze the memory requirements of each layer in the vision and language models individually and adaptively determine the optimal re-computation strategy for each layer.

**Balanced Adaptive Re-Computation Strategy**. In this context, $Q_v$ and $Q_t$ represent the inputs for Vision Transformer (VIT) and Large Language Model (LLM), respectively. $M_r$ denotes the remaining GPU memory at the current stage, while $M_t$ and $M_v$ indicate the GPU memory saved by each transformer layer of the LLM and VIT when enabling re-computation.

**Step-1**: Given the inputs $Q_v$ and $Q_t$, we enable the re-computation strategy across all transformer modules of the

*Table 2.* Main Results. We use open-source InternVL-Chat-1.5 6+20B and 6+34B as the models with either DeepSpeed (ZeRO-3) or Megatron-Deepspeed backend. GPU Days are reported in the InvernVL-Chat-1.2 1.2M training dataset to show the speed-up ratio. Models are also evaluated on five commonly used benchmarks.

| Model | Balance? | Backend | MMB-EN/CN | ChartQA | AI2D | MMVet | MME | GPU Days (speed-up) |
|---|---|---|---|---|---|---|---|---|
| 6+20B | × | DeepSpeed | 78.2/77.4 | 86.2 | 71.3 | 48.9 | 1901.2 | 38.9 (1.00×) |
| | ✓ | DeepSpeed | 78.7/77.6 | 86.5 | 71.4 | 50 | 1969.4 | 25.3 (**1.54×**) |
| | × | Megatron | 79.5/77.7 | 87.3 | 71.6 | 45.0 | 1957.7 | 61.8 (0.63×) |
| | ✓ | Megatron | 78.6/77.5 | 86.7 | 70.9 | 48.5 | 1956.3 | 21.3 (**1.83×**) |
| 6+34B | × | DeepSpeed | 80.0/79.2 | 86.6 | 73.4 | 45.9 | 2015.8 | 54.3 (1.00×) |
| | ✓ | DeepSpeed | 80.9/79.0 | 89.1 | 73.3 | 47.0 | 2153.6 | 35.5 (**1.53×**) |
| | × | Megatron | 80.2/79.3 | 88.9 | 73.7 | 44.2 | 2111.9 | 75.4 (0.72×) |
| | ✓ | Megatron | 80.1/78.0 | 89.3 | 73.5 | 45.4 | 2072.7 | 30.5 (**1.80×**) |

model. At each forward pass, we clear the cache and record each stage's remaining memory usage $M_r$.

**Step-2**: We manually disable re-computation for some layers based on the remaining GPU memory. Subsequently, we record the GPU memory usage $M_r'$ for each stage.

**Step-3**: Based on the memory differences $\Delta M_r$ observed between Step-1 and Step-2, along with the re-computation strategy implemented at each stage, we estimate the memory savings $M_t$ and $M_v$ for each transformer layer of the VIT and LLM, respectively.

**Step-4**: Based on the estimated GPU memory savings $M_t$ and $M_v$ measured in Step-3, as well as the remaining memory $M_r$ from Step-1, we first estimate the theoretically optimal re-computation strategy for each stage and conduct the training test. If the test runs successfully, we adopt this strategy. If it fails, we incrementally increase the number of re-computation layers by the remaining GPU memory for each stage.

# 5. Experiments

In this section, the models and datasets are introduced. Then, we demonstrate the acceleration compared to the current state-of-the-art VLMs. Subsequently, a detailed comparison of each component proposed in our method is presented, highlighting its specific contribution to training acceleration. Finally, extensive experimental analysis is conducted.

## 5.1. Experimental setup

**Model & Dataset setting:** We conduct experiments following the open-source InternVL-Chat-1.5 setting. Our vision and language models are InternViT-6B and InternLM2-20B, respectively. Two configurations are employed: InternVL-Chat-1.5 (6+20B) and InternVL-Chat-1.5-Plus (6+34B). As the InternVL-Chat-1.5 dataset is not yet available, we utilize the InternVL-Chat-1.2 dataset, which comprises approximately 1.2 million samples, as an alternative. All other training settings remain unchanged. GPU Days are our

evaluation metric to estimate the total training time. Specifically, GPU Days are reported based on A100 GPU usage to evaluate the speed-up performance.

## 5.2. Implementation Details

**How to get $Q_v$ and $Q_t$.** We determine $Q_v$ and $Q_t$ using dataset statistics. The total text token lengths and image count are used to compute the average tokens per image. $Q_t$ is set to the longest text token length, and the text-to-image ratio determines $Q_v$. It is challenging to maintain an exact number of images and text length, so we relax these conditions to allow for approximation. For images, $Q_v' = Q_v$, and for text, $Q_t' = Q_t - 128$ based on results in Table 6. In the InternVL-Chat-1.2 dataset, $Q_t = 4K$, $Q_v = 9$, with each image processed into 1K tokens for VIT.

**Sampling Overhead**. The sampling overhead is minimal. For instance, with 1.2 million samples, sampling can be completed in just a few tens of seconds, adding negligible overhead to the overall training process. The time complexity of ISF is $\mathcal{O}(C \cdot (N + M))$, where $N$ is the total number of samples, $M$ is the number of samples per pack, and $C$ is the number of iterations.

**Partition Overhead**. Establishing the anchor requires profiling layer-wise forward time and activation values, which takes only 5 steps and incurs minimal overhead. While the anchor may not be optimal, we observe empirically that the true optimum typically lies within $r \leq 3$ of it.

**Optimal Solution Overhead**. To refine the anchor, we measure the speed of the top 10–15 nearby partitions (as in Section 4.2), each requiring just 3 steps. This adds only a minor cost relative to the full training process.

## 5.3. Main Results

We demonstrate the superiority of OmniBal under various settings in Table 2. Baseline model is InternVL-Chat-1.5 (6+20B) (Chen et al., 2024), with DeepSpeed ZeRO-3 backend. OmniBal reduces GPU days from 38.9 to 25.3, achieving a 1.54× speed-up. Simultaneously, we maintain com-

*Table 3.* Importance of data balance. AVE-BS indicates average batch size. We report results with Model Balance (MB) and without MB.

| Method | AVE-BS | Max-Seq-Len | | Pad Ratio | Dist Ratio | | Balanced | GPU Days | |
|---|---|---|---|---|---|---|---|---|---|
| | | *VIT* | *LLM* | | *VIT* | *LLM* | | *w/o* MB | *w* MB |
| baseline | 4 | 20K | 16K | 0.31 | 0.34 | 0.30 | ✗ | 61.8 | 42.2 |
| length-group | 4 | 20K | 16K | 0.20 | 0.26 | 0.13 | ✗ | 54.0 | 40.0 |
| device-group | 4 | 20K | 16K | 0.378 | 0.125 | 0.15 | ✗ | 54.5 | 43.6 |
| **ISF(ours)** | 4.6 | **9K** | **4K** | **0** | **0.02** | **0.14** | ✓ | **51.9** | **29.0** |

*Table 4.* Importance of model balance. VAR indicates variance. SUM(comm) is the summation of commutation volume (MByte)

| Method | VAR(param) | VAR(num_layer) | VAR(fwd_time) | $\Delta$ SUM(comm) | GPU days |
|---|---|---|---|---|---|
| (1) parameter-based | **0.03** | 13.4 | 93.6 | +0.0 | 42.2 |
| (2) layer-based | 0.64 | **1.2** | 20.1 | +8.2 | 30.6 |
| (3) profile-based | 0.85 | 2.1 | **6.5** | +16.6 | 30.9 |
| (4) **BMP (ours)** | 0.83 | 1.5 | 12.2 | **-21.0** | **29.0** |

parable performance across commonly used datasets, such as MMB-EN/CN (Liu et al., 2023c), ChartQA (Masry et al., 2022), AI2D (Kembhavi et al., 2016), MMVet (Yu et al., 2023), and MME (Fu et al., 2023). Experiments with Megatron-DeepSpeed are conducted, which integrates tensor, pipeline, and data parallelism for larger-scale models. However, directly applying 3D parallelism can slow down training due to the heterogeneous nature of VLM models. Table 2 shows that switching to Megatron-DeepSpeed increased GPU days from 38.9 to 61.8. OmniBal addresses this issue by achieving computational balance across data, model, and memory, reducing GPU days from 61.8 to 21.3. This demonstrates the importance of computational balance for effective 3D parallelism. Notably, our method also outperformed DeepSpeed, highlighting the superiority of 3D parallelism when balanced computation is achieved. Results under a larger-scale setting (InternVL-Chat-1.5-Plus) are also reported to verify the generalizability of our method. The larger model consistently improves, accelerating the training process while maintaining model performance.

### 5.4. Ablation Analysis

In this section, ablation experiments on each component of our method are conducted, using InternVL-Chat-1.5 as the baseline model with a 3D parallel Megatron-DeepSpeed backend. Table 7 illustrates the impact of each component. The baseline model experiences a considerable slowdown in training speed due to computational imbalance, necessitating a total of 61.8 GPU days. By achieving data balance, GPU days are reduced to 51.9. Data balance allows us to achieve a more balanced model partition, reducing the training time. Finally, optimizing memory with an adaptive re-computation strategy reduces GPU days to 21.3. These results demonstrate that a holistic balance encompassing data, model, and memory is crucial for efficient VLM training. Below, we provide a detailed analysis of each component.

**The Importance of Data Balance:** In Table 3, we investigate the importance of data balance in large-scale distributed training by comparing four methods: (1) Baseline: Randomly combining data into a mini-batch with padding aligned to the longest input within mini-batches, (2) Length-Group: Combining samples with similar text and image sizes into a mini-batch to minimize padding within mini-batches. (3) Device-Group: Grouping samples with similar input sizes across devices to minimize idle times. (4) Balanced Dynamic Mini-batch: Using ISF to construct balanced mini-batches within mini-batches and across devices.

Table 3 reveals the following: (1) Baseline: is the slowest due to the completely random combination of different-sized samples, leading to significant size variation and excessive padding (0.31). Meanwhile, high Dist Ratio ViT (0.34) and LLM (0.30) result in computation disparities between devices, severely impacting throughput efficiency. (2) LengthGroup: enhances throughput efficiency by pre-grouping samples of similar sizes into mini-batches, thus reducing the internal padding ratio (0.2). Minimizing the number of redundant tokens within mini-batches effectively lowers the GPU days required to 54.0. (3) Device-Group: reduce idle time by ensuring consistent input sizes across devices. It decreases the Dist Ratio of ViT (0.125) and LLM (0.15). However, it only balances input sizes between devices and neglects the balance within mini-batches. High padding (0.378) wastes computational resources. (4) Our Approach: balances input sizes within mini-batches on each device and across devices simultaneously. It reduces both the Pad Ratio and the Dist Ratio, achieving a padding ratio of 0 while maintaining a lower Dist Ratio of 0.02 and 0.14. While our method balances input sizes, model partitioning still limits training speed. With model balance (MB), GPU days are reduced from 42.2 to 29.0, a gain of 13.2, compared to 9.9 without MB (from 61.8 to 51.9). This underscores the importance of a holistic balance approach.

*Table 5.* Importance of memory balance. VRAM$_i$ denotes remaining VRAM(G) in pipeline stage $S_i$. For the baseline model, the metric varies as <minimum> ~ <maximum>.

| Method | V-Seq-Len | L-Seq-Len | VRAM$_1$ | VRAM$_2$ | VRAM$_3$ | VRAM$_4$ | GPU Days |
|---|---|---|---|---|---|---|---|
| baseline | 4K~20K | 1K~16K | 13~50.2 | 7.3~40.5 | 7.3~40.5 | 7.3~40.5 | 61.8 |
| + data & model balance | 9K | 4K | 58.2 | 56.2 | 32.5 | 32.7 | 29.0 |
| + memory balance | 9K | 4K | 12.3 | 21.7 | 24.7 | 30.0 | **21.3** |

*Table 6.* Ablation results of $Q_v, Q_t$

| $Q_v$ | $Q_t$ | *DistRatio VIT* | *DistRatio LLM* |
|---|---|---|---|
| $Q_v$ | $Q_t - 64$ | 0.0159 | 0.145 |
| $Q_v$ | $Q_t - 128$ | **0.0156** | **0.144** |
| $Q_v$ | $Q_t - 256$ | 0.018 | 0.141 |
| $Q_v - 1$ | $Q_t - 128$ | 0.0417 | 0.20 |

*Table 7.* Ablation studies of components

| data balance | model balance | memory balance | GPU Days |
|---|---|---|---|
|  |  |  | 61.8 |
| ✓ |  |  | 51.9 |
| ✓ | ✓ |  | 29.0 |
| ✓ | ✓ | ✓ | **21.3** |

**The Importance of Model Balance:** Table 4 examines balanced model partitioning, focusing on partition strategies for pipeline parallelism. For LLM training, common methods include (1) parameter-based and (2) layer-based, (3) profile-based methods such as DreamPipe (Narayanan et al., 2019), which estimate the computation time for each layer, and use this information to partition the model effectively. Additionally, (4) our search-based Balanced Model Partition method finds the optimal partition strategy from a set of candidates. As shown in Table 4, (1) Parameter-based and (2) layer-based methods split the model's parameters or layers across devices, achieving low variation in VAR(param) and VAR(num_layer). However, they still show high variation in forward time VAR(fwd_time), leading to computational inefficiencies in the pipeline. (3) The profile-based method ensures the optimal VAR(fwd_time). However, this partitioning occurs before the vision model's token sub-sampling operation, increasing communication overhead and affecting training speed. (4) Our proposed BMP method explores a high-quality partition strategy space to identify the optimal strategy, achieving the best results in 29.0 GPU days.

**The Importance of Memory Balance:** In Table 5, we examine the significance of memory balance. In the baseline model, varying input sizes for vision (4K–20K tokens) and language (1K–16K tokens) lead to varying GPU memory usage. Despite aggressive re-computation, the remaining memory can drop to 7.3 GB. ISF and BMP improve training speed by controlling computational load across devices. However, memory demands still varied, *e.g.*, GPUs 1 and 2 having more remaining memory. ISF further speeds up

training by adjusting the re-computation strategy to fully utilize the remaining memory, reducing GPU days to 21.3.

**5.5. Generalization Capability**

We study the generalization capability of our method from multiple aspects: (1) Different Datasets like LLava-665K, InternVL-1.2M, and LCS558K in Appendix B; (2) Different Models: we verify our methods on different vision models like EVA-CLIP (Sun et al., 2023) and different language models like Llama3 (AI, 2024), Yi-34B (NousResearch, 2023), and the Qwen1.5-110B (Bai et al., 2023a), as detailed in Appendix C; (3) Different High-Resolution Setting: Under various settings, we achieved a speedup of approximately 2.0×, as demonstrated in Appendix D; (4) Different Tasks: Besides SFT tasks, pretraining tasks are also tested, as shown in Appendix E, and we observed consistent improvements across all settings; (5) Different Image Resolutions: As shown in Appendix F, our method consistently delivers a highly satisfactory acceleration effect with different input image resolutions; (6) Convergence of ISF: We show that ISF can converge in a few steps on other datasets, in Appendix G; (7) Different Model-series like LLava-1.6 in Appendix H; (8) Pre-Processing Strategy: Qwen2-VL (team, 2024) is a recent, highly regarded open-source project that provides strong support for dynamic image input. We adopt the pre-processing strategy of Qwen2-VL to validate our method in Appendix I; (9) Different Hardware like H100, A100 in Appendix J; (10) Larger-Scale Results. To validate our method, we conduct experiments with larger models and 512 GPUs. Results are provided in Appendix K. These results underscore the effectiveness and robustness of our method across a wide range of datasets, models, and tasks.

**6. Conclusion**

In this work, we effectively addressed the issue of imbalanced computation loads in large-scale 3D parallel training of vision-language models by rebalancing across data, model, and memory dimensions. Experimental results demonstrate that our method can significantly speed up training on many open-source models. The effectiveness and generalizability of our approach are also validated across various models, datasets, and hardware platforms. Our method can accelerate the development of this field by enabling more efficient training.

# Acknowledgements

This work was supported by the National Key Research and Development Program of China under Grant No. 2023YFB4405102. It was also supported in part by the Natural Science Foundation of Hunan Province under Grant No. 2024JJ6525. We would like to thank Kaihuan Liang, Shihao Bai, and Shuo Wu for their support on this work.

# Impact Statement

This paper presents work whose goal is to advance the field of Machine Learning. There are many potential societal consequences of our work, none of which we feel must be specifically highlighted here.

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

# A. Training-Pipeline of our balanced method

## A.1. Training-Pipeline

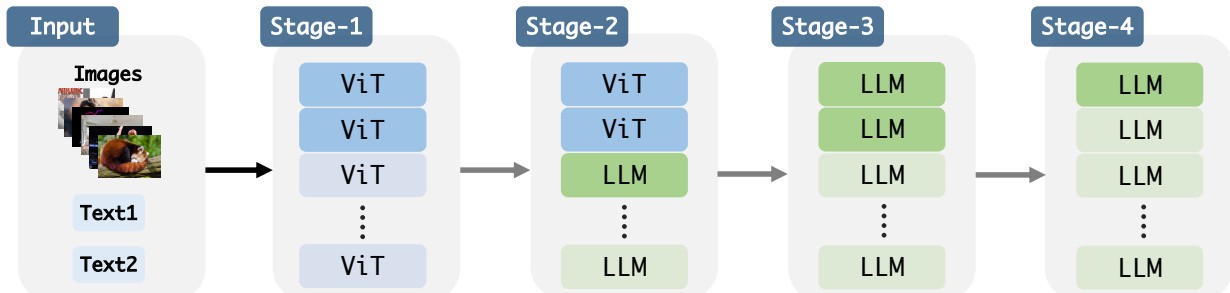

*Figure 3.* The pipeline consists of four stages, labeled Stage-1 to Stage-4, each representing a different stage of pipeline parallelism. Within this structure, "ViT" stands for the Vision Transformer layer, while "LLM" refers to the Transformer layer used for large language models (LLM). Regarding computational execution, the darker-colored sections signify forward passes with re-computation. In contrast, the lighter-colored sections denote a standard forward pass without re-computation.

# B. Results on different datasets

We consistently achieved a low Dist Ratio across the LLava-665K, InternVL-1.2M, and LCS558K datasets, as demonstrated in Table 8. Additionally, our approach significantly enhanced training speed.

*Table 8.* Results on different datasets

| Dataset | Dist Ratio | | GPU Days |
|---|---|---|---|
| | *VIT* | *LLM* | |
| LLava-665K | 0.02 | 0.145 | 43.3→ 12.4 (3.5×) |
| InternVL-1.2M | 0.02 | 0.14 | 61.8→ 21.3 (2.9×) |
| LCS-558K | 0.001 | 0.029 | 23.8→ 7.5 ( 3.2×) |

# C. Results on different Model Size

We test various combinations of vision and language models. As shown in Table 9, our approach significantly reduces the required GPU days for model training, achieving nearly a 2x speedup across models of various sizes.

# D. Results on Different Dynamic High-Resolution Settings

To validate the effectiveness of our method, we test it under various high-resolution settings. Our approach consistently demonstrates low Dist Ratio and strong acceleration across all configurations, as shown in Table 10, significantly improving training speed under different settings.

*Table 9.* Results for different model sizes are shown, with TP, PP, and DP representing various distributed training strategies: Tensor Parallel (TP), Pipeline Parallel (PP), and Data Parallel (DP), respectively. The "Stages-Layer-Num (V+L)" column indicates the number of Vision Transformer (V) and Language Transformer (L) layers assigned to each stage. Additionally, the "Re-computation" column denotes the number of re-computations enabled in each stage.

| Vision-Model | Language-Model | TP PP DP | Stages-Layer-Num (V+L) | Re-computation | GPU Days(speed up) |
|---|---|---|---|---|---|
| InternVL-6B | Llama3-8B | (1,4,8) | [16,17,20,24] | [8,7,10,24] | 27.7 → **13.8(2.0×)** |
| InternVL-6B | InternLM2-20B | (2,4,4) | [22,23,24,24] | [0,0,0,0] | 61.8 → **21.3(2.9×)** |
| InternVL-6B | Yi-34B | (4,4,2) | [28,29,24,24] | [3,2,0,0] | 75.4 → **30.5(2.5×)** |
| InternVL-6B | Llama3-70B | (4,8,2) | [22,23,13,14,14,14,13,12] | [11,12,8,5,3,2,0,0] | 129 → **52.5(2.4×)** |
| InternVL-6B | Qwen1.5-110B | (8,8,1) | [21,22,13,13,14,14,14,14] | [6,9,1,3,0,0,0,0] | 243 → **75.2(3.2×)** |
| EVA-CLIP-1B | InternLM2-20B | (2,4,4) | [43,16,15,14] | [0,0,0,0] | 23.6 → **12.2(1.9×)** |
| EVA-CLIP-4B | InternLM2-20B | (2,4,4) | [39,22,21,20] | [10,8,1,3] | 38.1 → **17.0(2.2×)** |
| EVA-CLIP-8B | InternLM2-20B | (2,4,4) | [17,18,23,22] | [5,5,8,10] | 41.8 → **20.3(2.0×)** |
| EVA-CLIP-18B | InternLM2-20B | (4,4,4) | [18,19,25,34] | [2,2,0,0] | 63.6 → **33.8(1.9×)** |

*Table 10.* Results on different dynamic high-resolution settings. "Max-Patch-Num" indicates the maximum number of patches into which an image can be divided. This parameter controls the granularity of image segmentation, impacting both model performance and computational efficiency. Adjusting the Max-Patch-Num allows for more flexible handling of high-resolution images in the model, optimizing resource usage while maintaining accuracy.

| Model | Max-Patch-Num | AVE-BS | Max-Seq-Len | | Dist Ratio | | GPU Days (speed-up) |
|---|---|---|---|---|---|---|---|
| | | | *VIT* | *LLM* | *VIT* | *LLM* | |
| InternVL-6B-20B | 1 | 7.6 | 9K | 4K | 0.06 | 0.05 | 28.6 → 13.7 (2.1×) |
| | 4 | 4.6 | 9K | 4K | 0.02 | 0.14 | 61.8→ 21.3 (2.9×) |
| | 6 | 2.7 | 14K | 5K | 0.019 | 0.136 | 147 →72 (2.05×) |
| | 12 | 1.9 | 14K | 5K | 0.03 | 0.12 | 209 →105 (2.0×) |

# E. Results on Pretrain Setting

We evaluate our method in other tasks, such as pre-training tasks.. In Pre-training, we train both the Vision Transformer (ViT) and MLP components for models ranging from 6B to 20B. However, for larger models, such as 6B-34B and 6B-70B, we focus solely on training the MLP component. Across all configurations, we observe consistent performance improvements shown in Table 11, particularly with the largest model, where GPU days are significantly reduced from 16.8 to 9.6, demonstrating enhanced training efficiency.

*Table 11.* Results on Pretrain Setting

| Model | Dataset | Trainable Module | AVE-BS | Dist Ratio | | GPU Days |
|---|---|---|---|---|---|---|
| | | | | *VIT* | *LLM* | |
| InternVL-6B-20B | LCS-558K | ViT+MLP | 5.8 | 0 | 0.03 | 9.9 → 6.0 (1.65×) |
| InternVL-6B-34B | LCS-558K | MLP | 5.1 | 0 | 0.031 | 8.3 → 4.9 (1.69×) |
| InternVL-6B-70B | LCS-558K | MLP | 5.2 | 0 | 0.029 | 16.8 → 9.6 (1.75×) |

# F. Results on Different Resolutions

We further test our method with different image resolution inputs. As shown in Table 12, our method consistently delivers low Dist Ratio and highly satisfactory acceleration results across varying image resolutions, demonstrating its effectiveness in improving training efficiency.

# G. Convergence of ISF.

The convergence performance of ISF is evaluated, with the results illustrated in Figure 4. On the LLava-665K dataset (Liu et al., 2023a), we observe that the Dist Ratio for both vision and language data dropped significantly after just one iteration.

*Table 12.* Results on Different Resolutions

| Resolution | AVE-BS | Dist Ratio | | GPU Days |
| --- | --- | --- | --- | --- |
| | | *VIT* | *LLM* | |
| 224 | 4.8 | 0.009 | 0.068 | $32.0 \rightarrow 20 \ (1.6\times)$ |
| 336 | 3.3 | 0.005 | 0.07 | $62.0 \rightarrow 33 \ (1.88\times)$ |
| 448 | 4.6 | 0.02 | 0.14 | $61.8 \rightarrow 21.3 \ (2.9\times)$ |

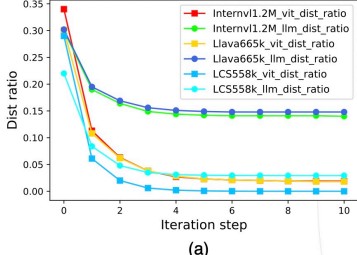 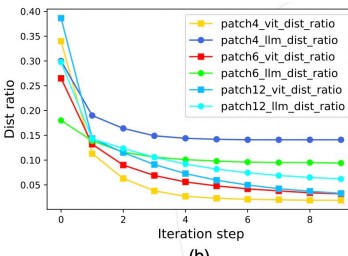 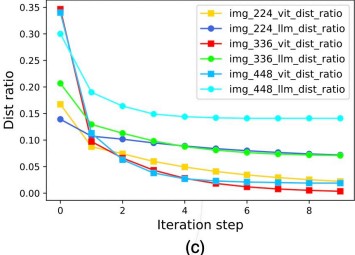

*Figure 4.* The convergence of ISF in various scenarios, including (a) different datasets, (b) different patch sizes, and (c) different image resolutions.

After five iterations, the Dist Ratio stabilized considerably. In practice, we perform ten iterations to ensure stable results, which only take less than one minute. The computational cost is negligible relative to the overall runtime. Additionally, our method is tested on two other datasets, InternVl-1.2M (Chen et al., 2024) and LCS558K (Liu et al., 2023b), and observed similar convergence rates.

## H. Results on Open-source LLava-1.6

We also validate our method using another popular open-source model, LLava-1.6, with the DeepSpeed backend, as shown in Table 13. For the DeepSpeed backend, we employ only our balanced dynamic mini-batch strategy. In the case of the open-source LLava model, while its ViT component is relatively small and the imbalance occurs primarily at the data level, we still achieved a notable overall speedup. Although the speedup ratio is smaller compared to other models, our method delivered a 30% improvement in performance.

*Table 13.* Results on Open-source LLava-1.6

| Model | AVE-BS | Dist Ratio | | GPU Days |
| --- | --- | --- | --- | --- |
| | | *VIT* | *LLM* | |
| Llava-1.6-7B | 4.54 | 0.008 | 0.037 | $10.2 \rightarrow 7.7 \ (1.3\times)$ |
| Llava-1.6-13B | 4.54 | 0.008 | 0.037 | $18 \rightarrow 13.3 \ (1.35\times)$ |
| Llava-1.6-34B | 4.4 | 0.009 | 0.0041 | $42.7 \rightarrow 31.3 \ (1.36\times)$ |

## I. Results on Qwen2-VL Pre-Processing Strategy

Qwen2-VL is a recent, highly regarded open-source project that provides strong support for dynamic image input. Consequently, we adopt the pre-processing strategy of Qwen2-VL to validate our method. As shown in Table 14, our approach demonstrates a substantial acceleration effect (approximately 1.9x) when applied to the Qwen2-VL strategy, significantly reducing both the padding ratio and dist ratio.

Table 14. Results on Qwen2-VL Pre-Processing strategy

| Model | Dataset | AVE-BS | Pad-Ratio | Dist Ratio | | GPU Days (speed-up) |
|---|---|---|---|---|---|---|
| | | | | *VIT* | *LLM* | |
| InternVL-6B-20B | InternVL-1.2M | 4 | 0.31 | 0.408 | 0.393 | 40.2 (1.0×) |
| InternVL-6B-20B | InternVL-1.2M | 6.6 | 0 | 0.12 | 0.06 | 21.0 (1.9×) |

## J. Different Hardware Results

We test our method on various hardware platforms with different GPUs (e.g., A100, H100) and network bandwidths. The experiments in Table 15 confirmed consistent performance improvements across all platforms.

Table 15. Results on Different Hardware. IB indicates network bandwidths.

| Dataset | Hardware | IB | Dist Ratio | | GPU Days (speed-up) |
|---|---|---|---|---|---|
| | | | *VIT* | *LLM* | |
| InternVL-1.2M | A100 | 4x200G | 0.02 | 0.145 | 61.8 → 21.3 (2.90×) |
| InternVL-1.2M | A100 | 2x200G | 0.02 | 0.145 | 64.0 → 24.8 (2.58×) |
| InternVL-1.2M | H100 | 8x400G | 0.02 | 0.145 | 32.5 → 12.2 (2.67×) |

## K. Large-Scale Results

To validate the effectiveness of our method, we conduct a study using larger-scale models and a greater number of GPUs. As shown in the Table 16, our method achieves a speedup ratio exceeding 2.0 across varying GPU configurations. Moreover, the results demonstrate that our approach maintains a more favorable linear speedup (85% → 95%) as GPUs increase.

Table 16. Results on Large-Scale models (6 + 70B) and GPUs

| Dataset | Hardware | IB | GPUs | Dist Ratio | | GPU Days (speed-up) |
|---|---|---|---|---|---|---|
| | | | | *VIT* | *LLM* | |
| InternVL-1.2M | H100 | 8x400G | 64 | 0.02 | 0.139 | 72.8 → 29.3 (2.48×) |
| InternVL-1.2M | H100 | 8x400G | 128 | 0.02 | 0.139 | 75.2 → 29.7 (2.53×) |
| InternVL-1.2M | H100 | 8x400G | 256 | 0.02 | 0.139 | 82.1 → 30.4 (2.70×) |
| InternVL-1.2M | H100 | 8x400G | 512 | 0.02 | 0.139 | 85.3 → 30.9 (2.76×) |

