# OpenReview forum: "OmniBal: Towards Fast Instruction-Tuning for Vision-Language Models via  Omniverse Computation Balance"
_ICML.cc/2025/Conference — ICML 2025 poster_

### Official Review · Reviewer_rnAP · 2025-02-20

**Overall Recommendation:** 3

**Summary:**

## update after rebuttal:
Score updated to 3

This paper suggest that during distributed training of vision-language models, there are computation imbalance due to model architectures, data types and how the mini batches are constructed inside and across devices. This paper proposes a novel method of mini-batch construction, model partitioning and re-computation of memory workload distribution so the computation imbalance is minimized, achieving a 1.8x faster training efficiency.

**Claims And Evidence:**

Not sure I can distinguish between evidence 1 (line 041) and evidence 3 (line 047)

**Essential References Not Discussed:**

None

**Experimental Designs Or Analyses:**

Experiments are well designed as per the table 3 and 4

**Methods And Evaluation Criteria:**

I think the methods and evaluation criteria are more empirical and not mathematical. The authors did a lots of experiments from different angles and they are easy to understand. But I do not see any mathematical formulation of their methods. Therefore it is harder to apply the techniques in other models. For example, the calculation of Qv and Qt is purely experimental. So if I want to find out those values for a new model, without any math formulation, I am left with running a bunch of experimentation

**Other Comments Or Suggestions:**

None

**Other Strengths And Weaknesses:**

Strength:
1. Well written paper with lots of analysis
2. The authors are tackling a novel problem of looking into efficient distributed training of vision-language models

Weakness
1. The paper lacks mathematical formulation so hard to use it in other models beyond the models shown in the experiment. For example if I want to do this for PaliGemma, how do I arrive at Qv and Qt without running a lot of experiments?

**Questions For Authors:**

1. How much overhead does the initial sampling stage adds?
2. How does it scales when you have more than 2 modalities - say language, audio and image? How will this impact the initial sampling overhead? Increase significantly or minimal?
3. I understand the inter-strage and intra-stage problem as described in figure 2. It would be good if you could show how this figure changes once you apply OnmiBal

**Relation To Broader Scientific Literature:**

this paper in on the broader topic of efficient distributed training such as LazyBatching (https://arxiv.org/abs/2010.13103) or DeepSpeed Zero etc. But the focus of this paper is on the vision-language model which adds a different flavor to existing works that are generally focused on single modality (specifically LLMs)

**Theoretical Claims:**

Table 7 shows ablation of Qv, Qt. But what is the mathematical formulation? Asking this because if I want to apply this technique to a new model, how do I find Qv and Qt for that?

---

> ### Author Rebuttal · Authors · 2025-03-30
>
> Thank you for your feedback.  Figures and tables are shown at https://anonymous.4open.science/r/O-A/O.pdf.
>
> *Q1:  Not sure I can distinguish between evidence 1 (line 041) and evidence 3 (line 047):*
>
>    "(1) Varying input sizes of LLM and VIT cause imbalance computational loads across training iterations and devices.
>
>    (3) Input size variation and computational imbalance compel us to use the most aggressive re-computation (checkpointing)  strategy to prevent program crashes, which wastes computational resources."
>
> **R1:**
> - (1) highlights the **idle time** across different devices caused by the variation in sequence lengths between the LLM and ViT, which leads to load imbalance during training.
> - (3) emphasizes the computational overhead introduced by the need for aggressive checkpointing strategies leading to **wasted computational resources**.
>
>
> *Q2： I think the methods and evaluation criteria are more empirical and not mathematical. The authors did a lots of experiments from different angles and they are easy to understand. But I do not see any mathematical formulation of their methods. Therefore it is harder to apply the techniques in other models. For example, the calculation of Qv and Qt is purely experimental. So if I want to find out those values for a new model, without any math formulation, I am left with running a bunch of experimentation.*
>
> Concerns about the mathematical formulation of methods and how to find Qv and Qt.
>
> **A2:**
>
> **Why not a mathematical formula?**
>
> The core of ensuring data balance lies in simultaneously fixing both the ViT input and the LLM input, which constitutes a Two-Dimensional Balanced Partition Problem (2D-BPP), an **NP-hard problem**, making it difficult to derive a general mathematical formulation. Instead, we adopt a practical and intuitive approach using experiments from multiple angles to tackle the problem effectively.
>
> The imbalance arises from variable input lengths. Our goal is to approximate equal input lengths for ViT and LLM, but no explicit optimal solution exists. Thus, we introduce Q'v and Q't as dataset-dependent hyperparameters to guide the balancing process.
>
> **How to get Q'v and Q't?**
>
> We provide a script (https://anonymous.4open.science/r/omnibal_example-E4D7/test_balanced_dynamic_batch.py, line 227) that uses an offline exhaustive search (≈10 minutes for 1.2M dataset only once, **no full training required**) to automatically determine Q'v and Q't, making the method easy to apply in practice.
>
> *Q3: How much overhead does the initial sampling stage adds?*
>
> **A3:**
>
> **Overhead Analysis**
>
> The sampling overhead is very low for example, with 1.2 million samples, the sampling can be completed within a **few tens of seconds**, which imposes minimal overhead on the overall training process.
>
> The time **complexity of ISF** is C*O(N+M), where N is the number of samples, M is the number of samples per pack and C is the number of iterations.
>
> *Q4:  How does it scales when you have more than 2 modalities - say language, audio and image? How will this impact the initial sampling overhead? Increase significantly or minimal?*
>
> **A4:**
>
> **Scaling to More Than Two Modalities**
>
> We begin by categorizing multi-modal tasks using the case of three modalities: language, audio, and image. When a task involves only two modalities (e.g., **ViT + LLM, LLM + Audio, or Audio + ViT**), our method can be directly applied with minimal modifications. These cases resemble standard VLM scenarios and are illustrated in the upper part of Figure 3  (https://anonymous.4open.science/r/O-A/O.pdf).
>
> For tasks involving all three modalities (**ViT + LLM + Audio**), we also provide adapted examples in the lower part of Figure 3 (https://anonymous.4open.science/r/O-A/O.pdf). In these cases, the underlying principles of our approach remain consistent. The only change is the need to handle more combinations during batch construction.
>
> **How will this impact the initial sampling overhead?**
>
>  Increase minimal for our ISF.  The time complexity of ISF is C*O(N+M),  which is essentially an **O(N)-level** method and therefore does not introduce significant overhead.
>
> *Q5: I understand the inter-strage and intra-stage problem as described in figure 2. It would be good if you could show how this figure changes once you apply OnmiBal*
>
> Figure 4 (https://anonymous.4open.science/r/O-A/O.pdf) shows the results of using Omnibal.
>
> Feel free for any other comments.

---

> > ### Comment · Reviewer_rnAP · 2025-04-04
> >
> > Thank you for your answers to my questions. I will increase my score to 3

---

> > > ### Author Response · Authors · 2025-04-06
> > >
> > > Thank you very much for your thoughtful response and for kindly considering increasing the score to 3.
> > >
> > > Just as a gentle note, the score may not have been updated yet on the system interface—perhaps it was just overlooked.

---

### Official Review · Reviewer_NXGX · 2025-03-12

**Overall Recommendation:** 4

**Summary:**

This work focuses on addressing imbalanced computational loads in large-scale 3D parallel training of vision-language models by rebalancing across data, model, and memory dimensions. The authors conduct experiments on various models, datasets, and hardware platforms to demonstrate the speed-up ratio for vision-language model training.

**Claims And Evidence:**

1. The claim that "vision-language instruct-tuning models have recently made significant progress due to their more comprehensive understanding of the world" is unclear. The statement implies that the progress is caused by a deeper understanding of the world, yet the authors provide no evidence or explanation for how such understanding is achieved or how it leads to the observed improvements.

2. Regarding the heterogeneity between LLM and ViT models, while this disparity is an objective fact, when the two models are fine-tuned as a unified system, it is unclear why one should consider the structure of each module separately. The authors fail to explain in detail what consequences arise from this heterogeneity—for example, how imbalanced model partitioning is affected and how the partitioning is specifically conducted.

3. The paper focuses on the instruction-tuning stage, but pretraining is equally important for building a strong multimodal LLM (MLLM). The authors do not mention pretraining or explain why it is not addressed in this work.

**Essential References Not Discussed:**

N/A

**Experimental Designs Or Analyses:**

1. The experiments first verify, across different backends and model sizes, that the proposed method not only speeds up training but also maintains training effectiveness.
2. Subsequently, the authors separately evaluate improvements from the data, model, and memory perspectives.

**Methods And Evaluation Criteria:**

1. The authors propose to enhance training speed from three aspects: balanced dynamic mini-batches, balanced model partition, and balanced adaptive re-computation.
2. They validate the efficiency improvements of their method across different backends and balancing strategies.

**Other Comments Or Suggestions:**

The font in Figures 3 and 4 is hard to read and should be improved for clarity.

If the authors address my concerns, I will raise the score.

**Other Strengths And Weaknesses:**

N/A

**Questions For Authors:**

1. Why does using data balancing affect the maximum sequence length, and how is this reduction achieved?
2. The authors mention that existing vision-language models suffer from issues such as structural heterogeneity and varied input sizes, which lead to slower training speeds. However, if future MLLMs adopt an architecture similar to the "fuyu" model—directly inputting visual tokens into the LLM without a separate visual encoder—would these issues persist? Would the proposed method still be effective under such circumstances?

**Relation To Broader Scientific Literature:**

1. The authors propose an iterative sampling and filtering strategy to improve data balance.
2. They utilize the sum of VAR (forward time) and communication time as a metric for partition evaluation, ranking the top K candidates for speed assessment.
3. Finally, the re-computation strategy is optimized based on actual memory needs.
4. The combined improvements across these three aspects lead to a significant increase in training speed, which is beneficial for the faster construction of MLLMs.

**Theoretical Claims:**

No significant theoretical claims are made.

---

> ### Author Rebuttal · Authors · 2025-03-30
>
> Thank you for your feedback.  Figures and tables are shown at https://anonymous.4open.science/r/O-A/O.pdf.
>
> *Q1: The claim that "vision-language instruct-tuning models have recently made significant progress due to their more comprehensive understanding of the world" is unclear. The statement implies that the progress is caused by a deeper understanding of the world, yet the authors provide no evidence or explanation for how such understanding is achieved or how it leads to the observed improvements.*
>
> **A1:**  We acknowledge that the original explanation may lack clarity. In the revised version, we will provide a more rigorous and precise description, clarifying the mechanisms behind the improvements in vision-language instruct-tuning models and avoiding ambiguous references to "understanding of the world.
>
> *Q2: Regarding the heterogeneity between LLM and ViT models, while this disparity is an objective fact, when the two models are fine-tuned as a unified system, it is unclear why one should consider the structure of each module separately. The authors fail to explain in detail what consequences arise from this heterogeneity—for example, how imbalanced model partitioning is affected and how the partitioning is specifically conducted.*
>
> **A2:**
>
> **Reasons for considering the structure of each module separately**
>
>  - **Input differences:** ViT and LLM modules receive fundamentally different types of inputs with distinct length distributions.
>  - **Architectural differences:** Although both ViT and LLM are based on Transformer architectures, their configurations differ significantly, resulting in unequal computational overhead.
>
> **Consequences arising from this heterogeneity**
>  - **Data-Level:** As shown in Figure 1 (https://anonymous.4open.science/r/O-A/O.pdf), when using LLM-style simple packing in a vision-language system, the ViT component suffers from a data imbalance problem.
>  - **Model-Level:** Figure 5 (https://anonymous.4open.science/r/O-A/O.pdf) shows the simple example model partition based on LLM's method. Since computational cost depends on both the architecture and input length, the architectural and input discrepancies between ViT and LLM make it difficult to evenly divide computation across pipeline stages. This results in an imbalanced workload and significant pipeline bubbles when applying traditional pipeline parallelism strategies designed for LLMs.
>
> *Q3. The paper focuses on the instruction-tuning stage, but pretraining is equally important for building a strong multimodal LLM (MLLM). The authors do not mention pretraining or explain why it is not addressed in this work.*
>
> **A3:**   We have also conducted experiments on multimodal large model pretraining. Details can be found in Section 5.4(4), Generalization Capability, under "Different Tasks," where we state: "Besides SFT tasks, pretraining tasks are also tested, as shown in Appendix G, and we observed consistent improvements across all settings." These results demonstrate that our approach remains effective even in the pretraining stage.
>
> *Q4: Why does using data balancing affect the maximum sequence length, and how is this reduction achieved?*
>
> **A4:**
>
> Figure 2 (https://anonymous.4open.science/r/O-A/O.pdf) shows details examples of how this reduction is achieved.
>
> **Simple Fixed Batching**
>
> The "maximum length" here refers to the max-seq-len in Table 5, including the batch dimension. The maximum input length is 5K tokens for the ViT and 4K tokens for the LLM. And with a batch size of 4 (as in InternVL-1.5), this yields 20K for ViT and 16K for LLM.
>
> **ISF Dynamic Batching**
>
>  Since ISF adopts a dynamic batching strategy, it can flexibly manage sequence lengths.  If some samples are relatively long, the corresponding batch size will be set smaller to keep the maximum sequence length (VIT 9K and LLM 4K).
>
> *Q5: The authors mention that existing vision-language models suffer from issues such as structural heterogeneity and varied input sizes, which lead to slower training speeds. However, if future MLLMs adopt an architecture similar to the "fuyu" model—directly inputting visual tokens into the LLM without a separate visual encoder—would these issues persist? Would the proposed method still be effective under such circumstances?*
>
> **A5:**
>
> "fuyu" model—directly inputting visual tokens into the LLM without a separate visual encoder making the model behave like LLM. The structural heterogeneity and input size imbalance issues will **disappear**.
>
> However,  its performance still lags behind more established architectures such as ViT-MLP-LLM. Therefore, our proposed method remains highly relevant to the current mainstream vision-language models.
>
> *Q6: The font in Figures 3 and 4 is hard to read and should be improved for clarity.*
>
> **A6:** We sincerely appreciate the reviewer’s valuable feedback and will update it in the revised version of the paper.
>
> Feel free for any other comments.

---

> > ### Comment · Reviewer_NXGX · 2025-04-02
> >
> > Thanks for the authors' rebuttal, which addresses all my concerns. I keep my score as 4.

---

> > > ### Author Response · Authors · 2025-04-06
> > >
> > > Thank you for your feedback. We sincerely appreciate your comments and response.

---

### Official Review · Reviewer_vzDC · 2025-03-17

**Overall Recommendation:** 3

**Summary:**

This paper identifies a significant computational imbalance issue in large-scale distributed training for Vision-Language Models (VLMs) due to heterogeneity in vision and language components. To tackle this, the authors propose OmniBal, a comprehensive framework that balances computation across three dimensions—data, model, and memory—by adaptively forming balanced mini-batches, optimizing model partitioning, and strategically adjusting memory re-computation. Extensive experiments demonstrate that OmniBal significantly reduces training time while maintaining accuracy, confirming its efficacy and versatility across various models, datasets, and hardware configurations.

**Claims And Evidence:**

Please refer to the Question.

**Essential References Not Discussed:**

Please refer to the Question.

**Experimental Designs Or Analyses:**

Please refer to the Question.

**Methods And Evaluation Criteria:**

Please refer to the Question.

**Other Comments Or Suggestions:**

Please refer to the Question.

**Other Strengths And Weaknesses:**

Strength:

1. The paper tackles an important challenge of computational imbalance in large-scale distributed training of vision-language models (VLMs).
2. It introduces a comprehensive framework that systematically addresses this imbalance from three complementary angles: data input balancing, optimized model partitioning, and adaptive memory management.
3. The motivation/observation of the imbalanced computation in VLMs is clear.

Weakness:
1. Some claims need a more detailed explanation.
2. The novelty of this paper needs further justification.
3. Some design choices seem ad-hoc and need justification.

Please refer to the Questions section for details.

**Questions For Authors:**

Thanks for the submission to ICML. I have the following comments/questions on this submission:

&nbsp;
1. The data partitioning approach in this paper appears to offer limited novelty. Numerous adaptive data batching methods, such as [1] and [2], have already addressed similar issues, albeit without specific evaluations on VLMs. However, applying these techniques to VLMs does not seem particularly challenging, given that (i) the primary goal—balancing computation both within and across batches—is quite similar, and (ii) the methodology, which involves batching based on profiled or predicted latency, closely resembles the existing works. Could the authors clarify the unique contributions of the proposed data partitioning method in comparison to these established batching approaches?

In otehr words, what are the major challenges that prevent applying those approaches to VLMs? If it is a simple adaptation, I have concerns on the contribution of this paper.

2. Section 4 introduces Q'v and Q't to reduce Pad Ratio and Dist Ratio, but the paper lacks a clear formula or explanation of how these values affect the ratios. This makes it challenging to understand how Q values are chosen to achieve the intended balance.
3. In Section 5, Q′v is set equal to Qv, while Q′t is defined as Qt − 128. Could the authors provide an intuition or rationale behind these choices?
4. The design of data balancing seems a little bit ad-hoc. It seems like a trial-and-error method that relies heavily on some empirically picked hyper-parameters. Consequently, it's unclear whether this method would generalize effectively to different evaluation datasets.

5. In the model partitioning design, it says that "A candidate set of partition strategies is created by jittering P(1), P(2), . . . , P(N−1) within a radius of r". Could the authors clarify how these parameters are chosen and whether the selection process is systematic or largely empirical?

6. I would like to see the comparison of the proposed method to some other computation balanacing frameworks such as Adapipe, besides showing the speedup over the baselines.

7. What is the difference between Figures 2 and 3?

**Relation To Broader Scientific Literature:**

Please refer to the Question.

**Theoretical Claims:**

N.A.

---

> ### Author Rebuttal · Authors · 2025-03-30
>
> Thank you for your feedback.  Figures and tables are shown at https://anonymous.4open.science/r/O-A/O.pdf.
>
> *Q1: what are the major challenges that prevent applying those approaches to VLMs?*:
>
> **Major Challenges:**
>
> - **Data Level:**
> Simple packing strategies for LLMs lead to a severe imbalance in the vision components when applied to VLMs shown in Figure 1 (https://anonymous.4open.science/r/O-A/O.pdf).
> - **Model Level:**
> Existing pipeline parallelism techniques assume homogeneous architectures for LLM and consistent input lengths. In VLMs, however, the ViT and LLM components differ significantly in both structure and input sequence length, leading to computational imbalance when such methods are naively applied.
>
> - **Memory Level:**
> The architectural mismatch between ViT and LLM also results in uneven memory consumption, rendering LLM-specific memory optimization strategies ineffective or even infeasible in VLM scenarios.
>
> **Illustrative Example:**
> Figure 1 illustrates a typical data imbalance issue that arises when applying simple LLM-style packing to VLM training.
>
> *Q2:  Section 4 introduces Q'v and Q't to reduce Pad Ratio and Dist Ratio, ... how Q values are chosen to achieve the intended balance.*
>
> **A2:**
>
> **Why not a mathematical formula?**
>
> The core of ensuring data balance lies in simultaneously fixing both the ViT input and the LLM input, which constitutes a Two-Dimensional Balanced Partition Problem (2D-BPP), an **NP-hard problem**, making it difficult to derive a general mathematical formulation.
>
> Our goal is to approximate equal input lengths for ViT and LLM, but no explicit optimal solution exists. Thus, we introduce Q'v and Q't as dataset-dependent hyperparameters to guide the balancing process.
>
> **How to get Q'v and Q't?**
>
> We provide a script (https://anonymous.4open.science/r/omnibal_example-E4D7/test_balanced_dynamic_batch.py,  line 227) that uses an offline exhaustive search (≈10 minutes for 1.2M dataset only once, **no full training required**) to automatically determine Q'v and Q't, making the method easy to apply in practice.
>
> *Q3: In Section 5, Q′v is set equal to Qv, while Q′t is defined as Qt − 128. Could the authors provide an intuition or rationale behind these choices?*
>
> **A3:** The choices of Q′v and Q′t are based on the search results discussed in **A2: How to get Q′v and Q′t?**. The goal is to minimize the DistRatio, and Appendix C provides an ablation table showcasing part of our search results supporting this decision.
>
> *Q4: The design of data balancing seems a little bit ad-hoc. It seems like a trial-and-error method that relies heavily on some empirically picked hyper-parameters. Consequently, it's unclear whether this method would generalize effectively to different evaluation datasets.*
>
> **A4:**
>
> **This is not ad-hoc**
>
> The data balancing strategy was carefully designed based on a thorough analysis of the data distribution and task requirements.
>
> **hyper-parameters**
>
> The parameters were also designed to solve the Two-Dimensional Balanced Partition Problem (2D-BPP), an **NP-hard problem**, and they are based on a search-based approach using dataset information, making them easily generalizable to other validation datasets.
>
> we conducted extensive experiments across diverse datasets, as detailed in Section 5.4 (**Generalization Capability**). These results demonstrate that our approach generalizes well beyond the initial evaluation setting.
>
>
> *Q5: In the model partitioning design, it says that "A candidate set of partition strategies is created by jittering P(1), P(2), . . . , P(N−1) within a radius of r". Could the authors clarify how these parameters are chosen and whether the selection process is systematic or largely empirical?*
>
> **A5:**
>
> **How these parameters are chosen**
>
> "P(1), P(2), ..., P(N−1) init" are obtained by profiling each layer’s forward time. A greedy algorithm computes the anchor partition strategy P⁺ to balance the computation time across all stages Si, which is **systematic**.
>
> "r" is **empirically** determined and spans the adjacent layers between the ViT and LLM to ensure a sufficient candidate space.
>
> *Q6: I would like to see the comparison of the proposed method to some other... Adapipe, besides showing the speedup over the baselines.*
>
> **A6:**
>
> **Comparison with other works:**
>
> Due to the limited number of studies specifically focused on large-scale VLM training, most existing approaches are designed for pure LLMs and cannot be directly applied.
>
> Model balance optimization in AdaPipe (combined with our data balance method ISF) is aligned with the profile-based method in Table 4 (paper) which achieves inferior results compared to our BMP.
>
> *Q7: What is the difference between Figures 2 and 3?*
>
> **A7:**  Same content, shown again for the reviewer to easily reference.
>
> Feel free for any other comments.

---

> > ### Comment · Reviewer_vzDC · 2025-04-04
> >
> > I would like to thank the authors for providing the additional results and clarifications. Although I still have some concerns regarding the novelty and potential overhead introduced by the trial-and-error parameter search design, I do not wish to be too picky since the experimental result is good.
> >
> > However, I still have concerns that applying methods such as [1, 2] to the VLM setting might be relatively straightforward and may not require substantial adaptation. While I acknowledge that VLM and LLM workloads and training paradigms are different, a deeper analysis beyond workload comparison is necessary—particularly at the design level. Of course, the original versions of these methods cannot be directly applied to VLM. But, for example, if we consider the data sizes of both the language and vision modalities during adaptive batching, is the adaptation relatively straightforward, or do significant challenges still remain?
> > I will increase my score if this concern is addressed.
> >
> >
> > [1] Yu, Gyeong-In, Joo Seong Jeong, Geon-Woo Kim, Soojeong Kim, and Byung-Gon Chun. "Orca: A distributed serving system for {Transformer-Based} generative models." In 16th USENIX Symposium on Operating Systems Design and Implementation (OSDI 22), pp. 521-538. 2022.
> >
> > [2] Choi, Yujeong, Yunseong Kim, and Minsoo Rhu. "Lazy batching: An sla-aware batching system for cloud machine learning inference." In 2021 IEEE International Symposium on High-Performance Computer Architecture (HPCA), pp. 493-506. IEEE, 2021.

---

> > > ### Author Response · Authors · 2025-04-06
> > >
> > > *Q: concerns that applying methods such as [1, 2] to the VLM setting might be relatively straightforward and may not require substantial adaptation.*
> > >
> > > **A:**
> > >
> > > **design level**
> > >
> > > Both [1] and [2] focus on batching for inference. Section 3 in [1] and Background Section B in [2]  highlight the differences between training and inference:  During training, dataset information is known in advance, whereas during inference, incoming requests are unpredictable. They mainly aim to optimize inference requests for serving, and there is a **clear gap** between batching in inference and training tasks.
> > >
> > >
> > > -  For **inference** [1, 2] refers to a technique in which incoming requests are dynamically grouped into batches in real-time, rather than waiting for a fixed-size batch, to improve throughput and balance latency and efficiency.
> > > -  For **training**, we need to ensure both efficient computation within each DP (Data Parallel) rank and workload balance across different DP ranks to minimize idle time.
> > >
> > > **More relevant work  in LLM Training**
> > >
> > > **Packing** enables efficient computation within each DP rank and balanced workloads across ranks. It is widely adopted by models like LLaMA, Qwen, and Deepseek, and frameworks such as **Megatron-LM, Nemo, and Huggingface Trainer**.
> > >
> > > https://docs.nvidia.com/nemo-framework/user-guide/24.07/nemotoolkit/multimodal/mllm/sequence_packing.html
> > >
> > > https://huggingface.co/docs/trl/v0.4.2/sft_trainer#packing-dataset-constantlengthdataset
> > >
> > > **LLM method (Packing) compared to VLM**
> > >
> > > - *Straightforward applied to VLM*
> > >
> > > Response A1 to Reviewer NPbH has shown the results. Packing in LLM results in computation imbalance problems (on VLM instruct-tuning).
> > >
> > > |   Model   |  Backend  |   Data Method  | Dist Ratio VIT | Dist Ratio LLM | GPU Days |
> > > |:---------:|:---------:|:--------------:|:---------------:|:---------------:|:--------:|
> > > |  6 + 20B  | Megatron  |     random     |      0.34       |      0.30       |  61.8    |
> > > |  6 + 20B  | Megatron  |  LLM packing   |      0.40       |      0.05       |  48.3    |
> > > |  6 + 20B  | Megatron  |      ISF       |      0.02       |0.14 | **21.3** |
> > >
> > >
> > > - *Challenge of transfer to VLM*
> > >
> > >
> > > The BFS (Best-fit-Shuffle) packing schemes in Megatron-LM already have high complexity O(N×M) (N: Number of samples; M: Number of samples per pack).
> > > If we further consider satisfying both LLM and ViT training simultaneously, it constitutes a Two-Dimensional Balanced Partition Problem (2D-BPP), which is **NP-hard**, making it difficult to derive a general mathematical formulation. Existing methods, when applied directly, fall short of addressing the specific challenges faced by our VLM.
> > >
> > > Our proposed heuristic solution, **ISF**, approximates the problem at **O(N)-level complexity** (as detailed in Response to Reviewer rnAP A3), effectively addressing the challenge. Moreover, it demonstrates a notable degree of adaptability in handling more than two modalities, as further elaborated in our response to Reviewer rnAP A4.
> > >
> > > We sincerely hope that our response helps address your concerns.

---

### Official Review · Reviewer_NPbH · 2025-03-17

**Overall Recommendation:** 3

**Summary:**

The paper addresses the causes of computational imbalance in VLM training, including aspects of data, model, and memory, and introduces OmniBal, a training framework designed for improving training efficiency of VLMs. OmniBal is basically comprised of three algorithms, balancing batch sizes, model partitions, and memory usages, respectively. The authors present experiments on various datasets and VLMs, claiming that their framework accelerates the process of VLM training under the metric of GPU days

**Claims And Evidence:**

Please see  Strengths and Weaknesses.

**Essential References Not Discussed:**

None

**Experimental Designs Or Analyses:**

Please see  Strengths and Weaknesses.

**Methods And Evaluation Criteria:**

Please see  Strengths And Weaknesses.

**Other Comments Or Suggestions:**

- There are some expressions and definitions that can be polished to be clearer for readers. In subsection 4.1, the exact term "Distribution ratio" should be placed right after the "Dist ratio" in bold. In subsection 4.2, when introducing the partitioning strategy, it might be better to use the form of the definition of partition in set theory. Both might make it clearer and avoid confusion.
- The explanation of Balanced Adaptive Re-Computation Strategy in Appendix B.2. can be moved to the main body to make the paper more organized.

**Other Strengths And Weaknesses:**

Strengths:
- This work demonstrates the detrimental impact of imbalanced data (batch size), distribution of model computation load, and re-computation on training efficiency of VLMs explicitly, showing thorough comprehension on the issue.
- Experiments across various datasets and VLM models verified the effectiveness of the OmniBal as a holistic framework, suggesting that it boosts training speeds by around 1.5 times to 3.5 times.

Weaknesses:
- In section 3 the paper mentions the differences between the training of VLMs and LLMs. However, it needs more solid explanations and explicit data to tell: (i) Whether present strategies experimented on LLMs can be transferred to VLMs? If they can, the data of their performance is expected. (ii) As mentioned in the same subsection, "(simple packing) results in computation imbalance problems (on VLM instruct-tuning)", then how severe the problems are?
- Effective as the experiments show, yet the solutions provided in the paper seem trivial. In addition, the details of optimizing re-computation in the Balanced Adaptive Re-Computation Strategy needs further explanation in Appendix B.1..
- In Table 4, the proposed method BMP only shows marginal improvement, compared with present methods.
- The experiments are not sufficient. (i) The paper demonstrates the effectiveness of OmniBal, but lacks comparisons between OmniBal as a holistic framework and other works on this issue. (ii) More ablation studies of different combinations of components are expected in Tabel 6, data + memory balance and model + memory balance for instance. Although these modules might perform better with prior ones applied, such experiments are still recommended.

**Questions For Authors:**

See Weaknesses.

**Relation To Broader Scientific Literature:**

None.

**Theoretical Claims:**

Please see  Strengths and Weaknesses.

---

> ### Author Rebuttal · Authors · 2025-03-30
>
> Thank you for your feedback.  Figures and tables are shown at https://anonymous.4open.science/r/O-A/O.pdf.
>
> *Q1:  In section 3 the paper mentions the differences between the training of VLMs and LLMs. However, it needs more solid explanations and explicit data to tell: (i) Whether present strategies experimented on LLMs can be transferred to VLMs? If they can, the data of their performance is expected. (ii) As mentioned in the same subsection, "(simple packing) results in computation imbalance problems (on VLM instruct-tuning)", then how severe the problems are?*
>
> **A1:**
>
> **Whether present strategies experimented on LLMs can be transferred to VLMs?**
>
> The **simple packing** strategy used in LLM training can be directly applied to VLMs, its effectiveness is limited due to the structural differences between the two. Figure 1 (available at https://anonymous.4open.science/r/O-A/O.pdf) provides an example illustrating how LLM-style packing manifests in VLM training.
>
> **(simple packing) results in computation imbalance problems (on VLM instruct-tuning) how severe the problems are?**
>
> The severity of the imbalance is demonstrated through actual analysis and experimental results, as shown in the accompanying table.
>
> |   model   | backend  | Data method | Dist Ratio VIT | Dist Ratio LLM | GPU Days |
> |:--------:|:--------:|:-----------:|:------------:|:------------:|:--------:|
> | 6 + 20B  | Megatron |   random    |     0.34     |     0.30     |   61.8 |
> | 6 + 20B  | Megatron | LLM packing |     0.40     |     0.05     |   48.3    |
> | 6 + 20B  | Megatron |     ISF     |     0.02     |     0.14     |   **21.3**   |
>
>
> *Q2:  Effective as the experiments show, yet the solutions provided in the paper seem trivial. In addition, the details of optimizing re-computation in the Balanced Adaptive Re-Computation Strategy needs further explanation in Appendix B.1.*
>
> **A2:**  Our problem is both  **challenging and important**, and to our knowledge, it has not been systematically studied before. Although our proposed solution is relatively simple, it is highly efficient and easy to transfer and apply.
>
> Regarding the optimization of re-computation in the Balanced Adaptive Re-Computation Strategy, we provided a brief explanation in Appendix B.1, as the underlying idea is relatively straightforward. We appreciate the reviewer’s suggestion and will include a more detailed discussion in a future version of the paper.
>
> *Q3: In Table 4, the proposed method BMP only shows marginal improvement, compared with present methods.*
>
> **A3:** BMP will yield more substantial benefits in more **communication-constrained** settings (larger models). BMP, compared to previous methods, takes into account the imbalance in point-to-point communication across different pipeline stages in VLMs.
>
> *Q4: the experiments are not sufficient. (i) The paper demonstrates the effectiveness of OmniBal, but lacks comparisons between OmniBal as a holistic framework and other works on this issue. (ii) More ablation studies of different combinations of components are expected in Tabel 6, data + memory balance and model + memory balance for instance. Although these modules might perform better with prior ones applied, such experiments are still recommended.*
>
> **A4:**
>
> **Comparison with other works:**
>
> Due to the limited number of studies specifically focused on large-scale VLM training, most existing approaches are designed for pure LLMs and cannot be directly applied.
>
> Model balance optimization in AdaPipe (combined with our data balance method ISF) is aligned with the profile-based method in Table 4 (paper) which achieves inferior results compared to our BMP.
>
> **Additional ablation studies:**
>
> The components in OmniBal are inherently interdependent and cannot be decoupled trivially.
>
> **Data-level balancing must be addressed first**, as it lays the foundation for subsequent optimizations at the model and memory levels.
>
> Moreover, memory optimization is intrinsically tied to model structure. As such, we did not perform isolated ablation studies. Instead, we adopted a progressive evaluation strategy, incrementally adding modules to assess their cumulative effectiveness.
>
> **Response to Other Comments Or Suggestions:**
>
> We sincerely appreciate the reviewer’s valuable feedback and will incorporate the suggested improvements in the revised version of the paper.
>
> Feel free for any other comments.

---

### Official Review · Reviewer_YLFx · 2025-03-19

**Overall Recommendation:** 3

**Summary:**

This paper focuses on the large-scale distributed training of multimodal large language models, and propose a omniverse computation strategy to manage vision-language data distribution and the training memory optimization.

Although the studied problem is significant in the development of multimodal large language models, the paper content does not really match the scope of ICML. For instance, the mainly compared work [Rajbhandari et al. 2020] is published at ICHPCNSA.

Moreover, the arguments made in this paper are often very subjectively. For instance, the authors claim that the data distribution affects distributed training efficiency. This statement has no any reference to support, and the experimental results are hard to make a direct connection with this point. Similarly, in the most part of this paper, the description lacks enough references to support, e.g., Sec.3. In the related work, there are also not other works mentioned about the distributed traing of VL models.

Besides, this paper seems to be finished in a harry. The presentation is very poor, making hard to really understand the motivation, methodology and contribution of this paper, although the experimental results seem significant.  The overall writing of this paper is more like a technical report, rather than an academic paper published by ICML.

Overall, I would like to suggest the authors to spend more time in polishing this paper, and then final a more relevant conference or journal for the submission.

**Claims And Evidence:**

See summary.

**Essential References Not Discussed:**

Yes

**Experimental Designs Or Analyses:**

Yes

**Methods And Evaluation Criteria:**

Yes.

**Other Comments Or Suggestions:**

See summary.

**Other Strengths And Weaknesses:**

See summary.

**Questions For Authors:**

See summary.

**Relation To Broader Scientific Literature:**

See summary.

**Theoretical Claims:**

Yes

---

> ### Author Rebuttal · Authors · 2025-03-30
>
> We thank the reviewer for the feedback.
>
> *Q1: ICML scope concerns*
>
> **A1:** We believe our paper is within the scope of ICML.
>
> **According to the ICML 2025 Call for Papers, topics of interest include (but are not limited to):
> "Machine Learning Systems (improved implementation and scalability, hardware, libraries, distributed methods, etc.)."**
>
> Our work focuses on large-scale distributed training and memory optimization for multimodal large language models, which fit the ICML scope.
>
> *Q2: in the most part of this paper, the description lacks enough references to support, e.g., Sec.3. In the related work, there are also not other works mentioned about the distributed traing of VL models*
>
> **A2:**
>
> In Section 2.1 Multi-Modal Large Language Models (MLLMs), we have cited a number of prior works on vision-language model (VLM) training, including **Qwen-VL, Q-Former, and LLaVA**. These papers, while primarily focused on model design and applications, do include brief discussions of distributed training strategies, often relying on backends such as DeepSpeed.
>
> Moreover, throughout the paper, we have provided extensive references to support our claims regarding distributed training and memory optimization, including established systems such as **PipeDream, DeepSpeed ZeRO, Megatron-LM, and GPipe**, among others.
>
> *Q3: Besides, this paper seems to be finished in a harry. The presentation is very poor, making hard to really understand the motivation, methodology and contribution of this paper, although the experimental results seem significant. "*
>
> **A3:**
>
>  While the current draft may need polishing in parts, the work was not done in haste. Other reviewers did not raise similar concerns about clarity.
> - *Reviewer rnAP comment "Well written paper with lots of analysis"*
> - *Reviewer vzDC comment "The motivation/observation of the imbalanced computation in VLMs is clear."*
>
> Feel free for any other comments.

---

> > ### Comment · Reviewer_YLFx · 2025-04-02
> >
> > Thanks for the authors' response. I carefully read the paper again, and saw more evidences in Appendix, which are ignored in the first review, so I upgrade my rating. As also commented by other reviewers, the main claims of this paper need more directly supports. I would like to suggest the authors to make a more clear comparison between the training of MLLM and LLM, rather than just leaving them in the Appendix. Besides, if the distributed training is within the scope of ICML, more relevant ICML references are suggested to add and discuss in the paper.

---

> > > ### Author Response · Authors · 2025-04-06
> > >
> > > Thank you very much for your valuable feedback. We will revise the paper according to the reviewers' suggestions by adding a detailed comparison between the training processes of MLLM and LLM in the main text and incorporating more relevant references.

---

### Decision · Program_Chairs · 2025-05-01

**Decision:**

Accept (poster)

**Comment:**

This paper proposes a large-scale 3D parallel training on the VLMs to solve the imbalanced computation load across different devices. The authors rebalance the computational load from data, model and memory perspectives, which achieves more balanced computation across devices. This paper received mixed reviews initially. The reviewers raised several issues including the reasonability of arguments, unclear paper presentation, unclear technique details, and insufficient experimental validations. During the rebuttal phase, the authors try to address these issues by providing more illustrations and results. These efforts have made reviewers unanimously lean towards positive afterward. Overall, the AC has checked all files, and stands on the reviewers' side. The authors shall incorporate relevant clarifications in the camera-ready version.